# Incidence of anorexia nervosa in young people in the UK and Ireland: a national surveillance study

Hristina Petkova,[1] Mima Simic,[2] Dasha Nicholls,[3] Tamsin Ford,[4] A Matthew Prina,[1] Ruth Stuart,[1] Nuala Livingstone,[5] Grace Kelly,[5] Geraldine Macdonald,[6] Ivan Eisler,[2] Simon Gowers,[7] Barbara M Barrett,[1] Sarah Byford [1]

For numbered affiliations see end of article.

**Correspondence to**
Professor Sarah Byford; s.byford@kcl.ac.uk

## ABSTRACT

**Objectives** This study aimed to estimate the incidence of DSM5 anorexia nervosa in young people in contact with child and adolescent mental health services in the UK and Ireland.

**Design** Observational, surveillance study, using the Child and Adolescent Psychiatry Surveillance System, involving monthly reporting by child and adolescent psychiatrists between 1st February 2015 and 30th September 2015.

**Setting** The study was based in the UK and Ireland.

**Participants** Clinician-reported data on young people aged 8–17 in contact with child and adolescent mental health services for a first episode of anorexia nervosa.

**Main outcome measures** Annual incidence rates (IRs) estimated as confirmed new cases per 100 000 population at risk.

**Results** 305 incident cases of anorexia nervosa were reported over the 8-month surveillance period and assessed as eligible for inclusion. The majority were young women (91%), from England (70%) and of white ethnicity (92%). Mean age was 14.6 years (±1.66) and mean percentage of median expected body mass index for age and sex was 83.23% (±10.99%). The overall IR, adjusted for missing data, was estimated to be 13.68 per 100 000 population (95% CI 12.88 to 14.52), with rates of 25.66 (95% CI 24.09 to 27.30) for young women and 2.28 (95% CI 1.84 to 2.79) for young men. Incidence increased steadily with age, peaking at 15 (57.77, 95% CI 50.41 to 65.90) for young women and 16 (5.14, 95% CI 3.20 to 7.83) for young men. Comparison with earlier estimates suggests IRs for children aged 12 and under have increased over the last 10 years.

**Conclusion** These results provide new estimates of the incidence of anorexia nervosa in young people. Service providers and commissioners should consider evidence to suggest an increase in incidence in younger children.

**Trial registration number** ISRCTN12676087.

## Strengths and limitations of this study

► The study benefits from a large, nationally representative sample from across the UK and Ireland.
► This study used a National surveillance system to collect data and thus avoided biases inherent in studying clinical samples via a small number of centres in a limited number of geographical areas.
► Results were limited by missing data which were dealt with by adjusting observed incidence rates using assumptions about incidence among missing cases.
► Results are relevant to young people diagnosed with anorexia nervosa by child and adolescent psychiatrists and not to those who are managed by general practitioners in primary care or those who have not come to the attention of services, for example, those who choose not to seek help.

100 000 for boys of the same age.[2] Prevalence estimates in young people range from 0.3% to 0.6%.[3 4]

Accurate epidemiological estimates of the number of new anorexia nervosa cases per year and their sex and age profile are needed for causal investigations and service planning.[2] However, available estimates in the UK are at least 10 years old.[2 5–7] In addition, most estimates are derived from community-based primary care records,[2 5] which fail to accurately record all new cases.[8 9] Undetected anorexia nervosa cases may present to accident and emergency and require immediate paediatric or psychiatric input, including inpatient admission. Some young people may therefore bypass primary care and, consistent with UK guidelines,[10] are likely to be assessed and diagnosed by a child and adolescent psychiatrist in a secondary care setting, making secondary care records a more reliable source of data on anorexia nervosa incidence than primary care registers.

Existing incidence data from secondary care settings in the UK are limited. One study,

## INTRODUCTION

Anorexia nervosa is a serious and enduring eating disorder with high morbidity and the highest mortality among psychiatric disorders.[1] Young women are particularly susceptible, with annual UK estimates of 37 new diagnoses of anorexia nervosa per 100 000 for girls aged 10–19 years, compared with 3 per

focusing only on adolescents aged 13 to 18, was limited to Greater London and reported presentation rates to child and adolescent mental health services (CAMHS), rather than incidence estimates.[6] A second study, which used a national surveillance design, focused only on children under 13.[7] The current study aimed to estimate the incidence of anorexia nervosa in secondary care services for young people between the ages of 8 and 17 years in the UK and Ireland. This work formed part of a study exploring the cost-effectiveness of models of care for young people with eating disorders (the CostED study).[11]

## METHODS
### Design
An observational, surveillance study was undertaken using the Child and Adolescent Psychiatry Surveillance System (CAPSS). CAPSS is a system designed to ascertain cases of rare childhood mental health conditions in the UK and Ireland through monthly reporting by clinicians and relies on non-consent to maximise the accuracy of epidemiological estimates. The CAPSS system has been operating since 2009[12] and is based on the well-established British Paediatric Surveillance Unit (BPSU) system.[13]

### Inclusion and exclusion criteria
The study included young people between 8 and 17 years of age, in contact with CAMHS for a first episode of anorexia nervosa according to DSM5 diagnostic criteria.[14] Anorexia nervosa is exceptionally rare in children under 8 and the cut-off at 17 was due to the focus on young people in contact with CAMHS, with many young people transitioning to adult services at the age of 18. New cases were notified for a period of 8 months from 1st February to 30th September 2015. Cases whose clinician-reported data were insufficient to assess eligibility were excluded, as were duplicate cases notified more than once by the same or different clinicians.

### Procedures
At the time of the study, CAPSS used a report card, known as the yellow card, containing a list of conditions being surveyed. Yellow cards, along with reporting instructions and protocols for new studies, are sent monthly from the CAPSS office to a mailing list of all hospital-, university- and community-based child and adolescent consultant psychiatrists across the UK and Ireland. Reporting clinicians are asked to check boxes against any of the reportable conditions they have seen in the preceding month, or to check a 'nil return' box and return the card to CAPSS. A tear-off slip is provided for respondents to keep a record of the patients reported. 'Positive' returns are allocated a unique CAPSS ID number and notified to the appropriate research investigator, who then contacts the reporting clinician directly to request completion of a questionnaire using the CAPSS ID to enable the clinician to identify the relevant patient.

For the CostED study, the yellow card contained a check box for anorexia nervosa and was sent to clinicians along with a protocol card detailing the case notification definition for anorexia nervosa. The case notification definition (see web extras) was based on DSM5 diagnostic criteria for anorexia nervosa and was intended to aid clinicians in their decision to tick 'yes' or 'no' on the yellow card. It was not intended to identify whether a case met study inclusion criteria, which was determined by the research group after receipt of all necessary data.

### Data
Questionnaires were sent to clinicians who reported a positive case of anorexia nervosa, identified via the unique CAPSS ID number. Questionnaires were completed from clinical records and clinicians were asked to provide data relating to the time the case was initially assessed and diagnosed. The questionnaire covered clinical features to enable assessment of case eligibility, referral pathway information to ensure assessment and diagnosis had not happened prior to the study surveillance period, and a limited set of standard patient identifiers in line with CAPSS procedures and ethics requirements, which were used to describe the sample and to identify duplicate notifications. In addition, clinicians were asked to confirm whether the case was a first episode of anorexia nervosa that had come to the attention of services.

The patient identifiers included NHS or Community Health Index (CHI) number (unique patient identifiers used in the regions of interest), hospital number, first half of postcode or town of residence for Ireland, sex, date of birth and ethnicity (white, mixed, Asian, black, Chinese, other or unknown). In Northern Ireland, identifiers were limited to age in years and months and hospital identifier rather than hospital number, to reduce the risk of patient identification given the small geographic area. All patient identifiable data from Northern Ireland were retained by the local research team, de-duplicated, anonymised and subsequently sent to the central research team in King's College London for analysis as per requirements set out by the Northern Ireland Privacy Advisory Committee. All data storage were compliant with the EU General Data Protection Regulations.

Clinical features included: weight and height to calculate body mass index (BMI) and percentage of median expected BMI for age and sex interpreted around the 85% threshold[15]; the Health of the Nation Outcome Scales for Children and Adolescents (HoNOSCA),[16] a routine outcome measure rating 13 clinical features on a five-point severity scale including behaviours, impairments, symptoms, and social functioning of children and adolescents with mental health problems; the clinician completed Children's Global Assessment Scale (CGAS)[17] used to rate emotional and behavioural functioning of young people; and a range of symptoms relating to the diagnosis of anorexia nervosa.

Unreturned or incomplete questionnaires were chased via email and telephone. Cases where any symptom

required for case definition was absent, despite chasing, were assessed for eligibility by a consultant child and adolescent psychiatrist (MS).

### Case eligibility

Cases were assessed as eligible for the study if: (1) they were between 8 and 17 years of age; (2) they had no previous episode of anorexia nervosa that had come to the attention of services; (3) they received a clinical assessment in the reporting service during the study surveillance period; (4) they had not been referred from another secondary health service (to ensure assessment and diagnosis had not happened prior to the study surveillance period); and (5) the following clinical symptoms were present: 'restriction of energy intake relative to requirements' and 'persistent behaviour that interferes with weight gain, despite low weight'. This broad definition was subsequently checked using a tighter DSM5 analytic definition including the following symptoms:

1. Restriction of energy intake relative to requirements.
2. Intense fear of gaining weight or becoming fat or persistent behaviour that interferes with weight gain, despite low weight.
3. Perception that body shape/size is larger than it is or preoccupation with body weight and shape or lack of recognition of the seriousness of the current low body weight.

Only one case which met the broad criteria failed to meet the tighter criteria, thus confirming the validity of the broad criteria.

### Removal of duplicates

Duplicates were identified by comparing NHS/CHI numbers, hospital numbers/hospital identifiers and date of birth/age in years and months, as appropriate. The management of duplicates depended on the outcome for the original notification for which a duplicate was identified. Four scenarios were considered: (1) duplicates where the original notification met study inclusion criteria were excluded and the original retained; (2) duplicates where the original notification had been excluded because the young person was under 8 years of age or did not meet the clinical criteria were assessed as a new case to determine if the case now met eligibility criteria; (3) duplicates where the original notification was excluded due to a previous episode of anorexia nervosa, a diagnosis date prior to the study surveillance period or referral from another secondary care service, were excluded and (4) duplicates where the original notification contained insufficient information to judge eligibility were checked to see if the duplicate contained the missing information and, if available, the original notification was reassessed for eligibility and the duplicate managed as per the scenarios above.

### Data analysis

Data analysis was performed using Stata IC V.14.2 and Microsoft Excel 2010. Observed incidence rates (IR0), defined as the number of new cases during a specified period of time in a population at risk for developing the disease, were calculated as follows: the number of confirmed new cases of anorexia nervosa in the 8-month surveillance period converted to 12 months [(N cases over 8 months/8) × 12], divided by the population at risk and multiplied by 100 000 to give the rate per 100 000 young people.

$$IR0 = \text{(confirmed new cases converted to 12 months)} / \text{the population at risk} \times 100\,000$$

The population at risk was calculated as the total number of children of each year of age and each sex in the UK and Ireland minus the number of prevalent cases who, once diagnosed, are no longer part of the 'at risk' population. Population data for 2015 were obtained from the Office for National Statistics for the UK[18] and the Central Statistics Office for Ireland.[19] To estimate the number of prevalent cases each year, incident cases in the previous age band were used as a proxy. For example, incident cases aged 8 were used as a proxy for prevalent cases in the estimation of the 'at risk' population aged 9 and so on.

To consider incidence among unobserved missing cases, adjustments were needed for unreturned CAPSS notification cards and questionnaires. For CAPSS notification cards, just over half (50.16%) of all notification cards sent out were returned. To account for incidence among the 49.84% of unreturned cards, two assumptions were made, and an appropriate correction applied to IR0, the observed IR:

*Assumption 1*: To take into consideration the possibility that unreturned cards are more likely to be 'nil' returns, it was assumed that half (24.92%) of unreturned cards were 'negative' and half followed the same proportion of 'negative' and 'positive' as the returned cards. This assumption translates into a correction coefficient of 1.50 derived from (24.92+50.16)/50.16.

*Assumption 2:* Making no assumptions of bias in the likelihood of unreturned cards being either positive or negative returns, it was assumed that all unreturned cards followed the same proportion of 'negative' and 'positive' as returned cards. This assumption translates into a correction coefficient of 1.99 derived from (49.84+50.16)/50.16.

These assumptions provide a range of IRs, from a minimum (observed IR) to a maximum (assumption 2), within which the actual rate is likely to fall. We hypothesised that assumption 1 provides the most realistic estimate since it assumes a bias in the response rates with greater likelihood that unreturned cards are negative ('nil' returns) but does not assume *all* unreturned cards are 'nil' returns, which is the implicit assumption within IR0.

For unreturned questionnaires, approximately two-thirds (63%) of the questionnaires that were sent to clinicians reporting positive cases of anorexia nervosa were returned, leaving one-third (37%) unreturned. Since all these questionnaires relate to a 'positive'

notification, we applied a correction coefficient of 1.59 derived from (37+63)/63, which assumes that the IR for the unreturned questionnaires is the same as the IR identified in the returned questionnaires for each year of age.

We then combined the correction coefficients described above, to generate two adjusted IRs:

Adjusted incidence rate 1 (IR1) = Confirmed new cases of anorexia nervosa converted to 12 months, multiplied by the correction for unreturned CAPSS notification cards under assumption 1, multiplied by the correction for unreturned questionnaires, then divided by the population at risk and multiplied by 100 000.

$$IR1 = (\text{confirmed new cases converted to 12 months} \times 1.50 \times 1.59) / \text{the population at risk} \times 100\,000$$

Adjusted Incidence rate 2 (IR2) = Confirmed new cases of anorexia nervosa converted to 12 months, multiplied by the correction for unreturned CAPSS notification cards under assumption 2, multiplied by the correction for unreturned questionnaires and then divided by the population at risk and multiplied by 100 000.

$$IR2 = (\text{confirmed new cases converted to 12 months} \times 1.99 \times 1.59) / \text{the population at risk} \times 100\,000$$

For each IR, IR0, IR1 and IR2, total, age-specific and sex-specific annual IRs for anorexia nervosa for the year 2015 and 95% CIs were calculated based on the Poisson distribution[20] using the Stata command ci means [N new anorexia nervosa cases 12 m], Poisson [exposure(-total population)] for positive integers/whole incidence numbers (Stata interprets any non-integer decimal point number between 0 and 1 as the fraction of events and converts it to an integer number). Annual IRs were stratified by discrete age and sex.

### Public and patient involvement statement
The CostED study included a patient and a parent representative on the study steering committee who contributed to the design, conduct and management of the study, including the incidence component.

## RESULTS
### Case ascertainment
Case ascertainment is outlined in figure 1. Over the 8-month surveillance period, 6401 yellow cards were sent to reporting clinicians and 3211 (50%) were returned. Of these, 997 positive cases of anorexia nervosa were reported and 2214 were nil returns. Of the positive cases, 48 (5%) were excluded due to clinicans stating that they did not wish to be included in the study (due to retirement, shortage of reporting capacity and so on) or due to reporting errors. Questionnaires were sent to the remaining 949, and a further 352 (37%) positive cases were excluded as they failed to return the questionnaires, so no data were available to assess eligibility. Questionnaires were completed and returned for 597 notified cases, of which 292 (49%) were ineligible for reasons

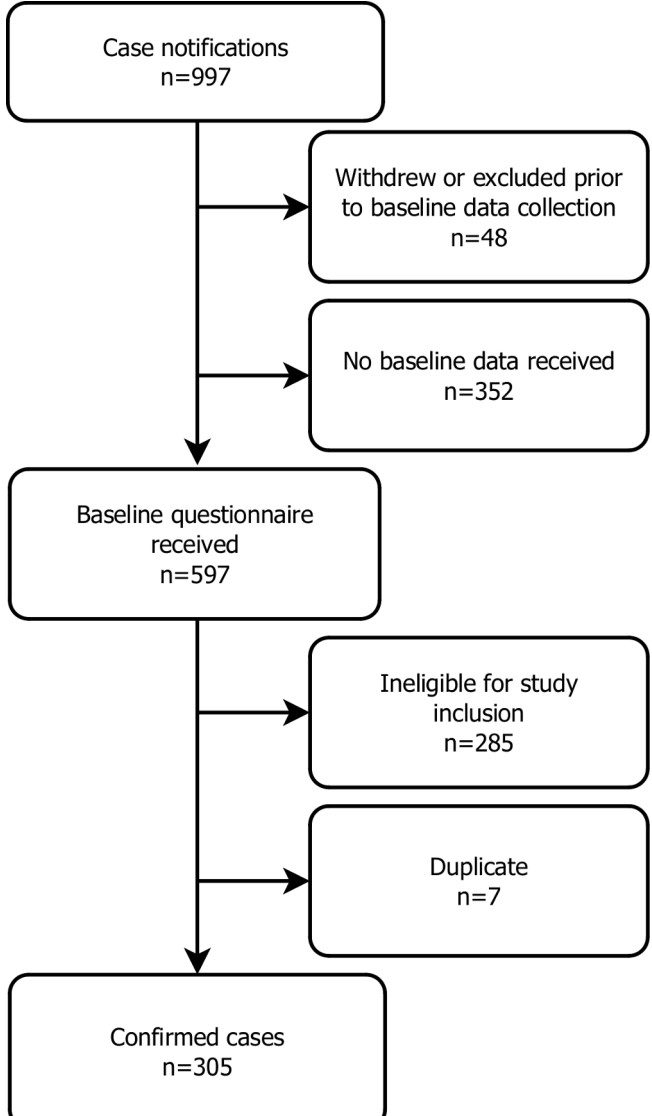

**Figure 1** Flow diagram of case ascertainment.

related to age, previous episode of anorexia nervosa, date of assessment outside the study's surveillance period, referral from another secondary care service, insufficient information to assess diagnosis or duplicate notifications, leaving 305 incident cases of anorexia nervosa as the sample for analysis.

### Demographics and clinical features
Of the 305 young people identified as having DSM5 anorexia nervosa, the majority (91%) were girls of white ethnicity (92%) (see table 1). The mean age was 14.6 years (±1.66). Clinical variables suggest these young people were significantly impaired. Mean BMI was 16.50 kg/m$^2$ (±2.25), where values of 16.00–16.99 suggest moderate severity of anorexia nervosa. Mean percentage of median expected BMI for age and sex (the deviation from expected body weight) was 83.23% (±10.99%), falling within the range required for a diagnosis of anorexia nervosa (<85%). Mean CGAS score was 44.61 (±14.08), which falls within the range for 'obvious

## Table 1 Characteristics of incident cases

| | N | Mean (SD) or % |
|---|---|---|
| Age | 305 | 14.56 (1.66) |
| Sex | | |
| Female | 279 | 91.48 |
| Male | 26 | 8.52 |
| Ethnicity | | |
| Any white | 274 | 91.64 |
| White and Asian | 6 | 2.01 |
| White and black Caribbean | 2 | 0.67 |
| White and black African | 1 | 0.33 |
| Other mixed | 1 | 0.33 |
| Indian | 3 | 1.00 |
| Pakistani | 2 | 0.67 |
| Bangladeshi | 1 | 0.33 |
| Other Asian | 4 | 1.34 |
| Black Caribbean | 2 | 0.67 |
| Chinese | 1 | 0.33 |
| Ethnicity not known | 2 | 0.67 |
| Clinical status | | |
| BMI | 304 | 16.50 (2.25) |
| %e of Median expected BMI | 303 | 83.23 (10.99) |
| CGAS | 280 | 44.61 (14.08) |
| HoNOSCA | 63 | 19.40 (8.17) |

BMI, body mass index; CGAS, Children's Global Assessment Scale; HoNOSCA, Health of the Nation Outcome Scales for Children and Adolescents.

problems' (41–50) on a scale from 1 to 100 (1 being the worst and 100 the best emotional and behavioural functioning). Mean total HoNOSCA score was 19.40 (±8.17) on a scale from 0 to 52, indicative of a severity similar to that at inpatient admission.[21 22]

The proportion of the included sample notified from each region within the British Isles is reported in table 2, alongside the population of young people in each region by age. England has the largest population (78%) and notified 70% of new cases. Scotland, containing only 7% of the total population, notified 14% of the sample and

Northern Ireland, containing only 3% of the population, notified 13% of the sample. By contrast, Ireland notified only 2% of cases, despite containing 8% of the population, and Wales notified no eligible cases (some cases were notified but did not meet inclusion criteria), despite containing 4% of the population.

### Incidence rates

Table 3 details $IR_0$s and adjusted IRs (IR1 and IR2) by age. IRs for the total sample ranged from a minimum of 5.75 per 100 000 young people (95% CI 5.23 to 6.30; IR0) to a maximum of 18.22 per 100 000 young people (95% CI 17.29 to 19.18; IR2), with IR1, the rate hypothesised to be the most accurate, falling between these two values at 13.68 per 100 000 population (95% CI 12.88 to 14.52). Focusing on IR1 rates, total incidence increased steadily with age, peaking at 16 (30.37, 95% CI 26.70 to 34.41), with a substantial drop at the age of 17 (14.35, 95% CI 11.88 to 17.19).

Table 4 reports IRs by age and sex. Incidence among young men followed a similar pattern to overall IRs reported in table 3, being highest at the age of 16 (5.14) and half that at age 17 (2.54). The highest incidence among young women was seen a year earlier than for boys, at the age of 15 (57.77), with similar rates at age 16 (56.95), dropping by more than half at age 17 (26.82).

### DISCUSSION

### Principal findings

This study provides new estimates of incident cases of anorexia nervosa in young people aged 8–17 presenting to CAMHS services in the UK and Ireland. Our mid-range, missing data-adjusted estimate (IR1) of the incidence of anorexia nervosa in the full sample of young people aged 8–17 years was approximately 14 per 100 000.

### Comparison with other studies

This result is lower than previous primary care-based estimates of 18–20 per 100 000 focusing on young people aged 10–19.[2 5] This difference is due to the different age ranges in the studies; the inclusion of children as young as 8 in the current study, who have relatively low incidence, and exclusion of adolescents aged 18–19, whose incidence is relatively high, makes the results difficult to

## Table 2 Cases by region of the UK and Ireland

| Region | Incident cases (N) | Incident cases (%)* | Population (N) | Population (%) |
|---|---|---|---|---|
| England | 213 | 69.84 | 6 194 444 | 77.83 |
| Scotland | 44 | 14.43 | 561 490 | 7.06 |
| Northern Ireland | 41 | 13.44 | 231 822 | 2.91 |
| Ireland | 7 | 2.30 | 628 251 | 7.89 |
| Wales | 0 | 0.00 | 342 627 | 4.31 |
| Total | 305 | | 7 958 634 | |

*Does not sum to 100 due to rounding.

**Table 3** Annual incidence of anorexia nervosa in young people aged 8–17 for 2015, reported per 100 000 young people

| Age | Observed rate IR0 | | Adjusted rate IR1 | | Adjusted rate IR2 | |
|---|---|---|---|---|---|---|
| | IR | 95% CI | IR | 95% CI | IR | 95% CI |
| 8 | 0.18 | 0.01 to 0.76 | 0.43 | 0.10 to 1.14 | 0.57 | 0.18 to 1.35 |
| 9 | 0.18 | 0.01 to 0.77 | 0.44 | 0.11 to 1.17 | 0.58 | 0.18 to 1.38 |
| 10 | 0.19 | 0.01 to 0.80 | 0.45 | 0.11 to 1.21 | 0.60 | 0.19 to 1.43 |
| 11 | 1.53 | 0.79 to 2.67 | 3.65 | 2.43 to 5.25 | 4.85 | 3.43 to 6.65 |
| 12 | 4.91 | 3.47 to 6.76 | 11.69 | 9.39 to 14.38 | 15.56 | 12.89 to 18.63 |
| 13 | 8.39 | 6.44 to 10.73 | 19.95 | 16.89 to 23.42 | 26.58 | 23.02 to 30.54 |
| 14 | 11.71 | 9.41 to 14.39 | 27.85 | 24.25 to 31.84 | 37.10 | 32.92 to 41.66 |
| 15 | 12.39 | 10.05 to 15.10 | 29.47 | 25.80 to 33.52 | 39.25 | 35.50 to 43.88 |
| 16 | 12.76 | 10.42 to 15.47 | 30.37 | 26.70 to 34.41 | 40.45 | 36.19 to 45.07 |
| 17 | 6.03 | 4.47 to 7.96 | 14.35 | 11.88 to 17.19 | 19.12 | 16.24 to 22.35 |
| Total | 5.75 | 5.23 to 6.30 | 13.68 | 12.88 to 14.52 | 18.22 | 17.29 to 19.18 |

IR, incidence rate.

compare. However, comparing rates for 10–14 year olds, available in the current study and in the most recent of these published estimates, produces similar incidence rates, with rates of 12.6 per 100 000 in the current study, compared with 13.1 per 100 000 in 2009.[2] For females, the rates are 23.3 per 100 000 in the current study compared with 24.0 per 100 000 in 2009 and for males, 2.4 and 2.5 per 100,000, respectively. However, this comparison should be treated with caution given the very different settings—primary care versus secondary care.

Existing secondary care estimates of the incidence of anorexia nervosa in the UK are limited to children under the age of 13, with an overall incidence of 1.09 per 100 000 reported for children aged between 6 and 12 between 2005 and 2006.[7] The methodology for this study was very similar to the CostED methodology, using the CAPSS system but additionally the BPSU System. For comparison with the current study, the incidence rate for children between 8 and 12 was approximately 1.5 per 100 000 for DSMIV anorexia nervosa or 2.1 per 100 000 for DSMIV anorexia nervosa plus those classified as 'other eating disorders' meeting CostED criteria for DSM5 anorexia nervosa (estimated from the original data by DN). This compares to a rate of 3.2 per 100 000 in the current study for children of the same age. This estimate is higher than both of the 2006 estimates suggesting that incidence rates for younger children have increased over time.

The results presented are also supported by international evidence. One study carried out in Italy demonstrated a significant reduction in age at onset for anorexia nervosa in consecutive outpatient referrals between 1985 and 2008 (n=1666).[23] A second study exploring time trends in the incidence of anorexia nervosa, which was carried out using data from the Norwegian National Patient Register, found overall rates of anorexia nervosa to be stable between 2010 and 2016 for the sample as a whole, but increasing for young females aged between 10 and 14.[24]

### Strengths and weaknesses of the study

The large, nationally representative sample of this study is a strength. The study included young people with anorexia nervosa, diagnosed using DSM5 criteria, from across the UK and Ireland and thus avoided biases inherent in studying clinical samples via a small number of centres in a limited number of geographical areas. The results are of relevance primarily to the UK and Ireland but may be of value to other high-income countries.

With only a 50% case notification response rate from CAPSS clinicians and a third of questionnaires not returned, missing data were a major constraint. There are many reasons why clinicians may fail to return notification cards or questionnaires, including changes in place of employment, competing priorities, or the belief that cases will be reported by a colleague.[13] This problem was addressed by adjusting the observed incidence rates using assumptions about incidence among both missing case notifications and missing questionnaires.

The methodology is also limited to young people seen by child and adolescent psychiatrists. Cases that would not be identified by this methodology include those who have not come to the attention of services, for example, those who choose not to seek help, those managed by general practitioners in primary care, and those in the care of mental health services without psychiatric input, such as nurse-led facilities. This latter concern was an issue in Northern Ireland where, due to initial low numbers of notifications, investigation by the research team identified a number of nurse-led facilities which were invited to contribute, and subsequently reported just over half of all cases in Northern Ireland. In terms of missing primary care cases, given UK guidelines for assessment and diagnosis of anorexia nervosa to be carried out by child and

**Table 4** Annual incidence of anorexia nervosa in young people aged 8–17 for 2015 by sex, reported per 100 000 young people

| Age | Observed incidence IR0 | | | | Adjusted incidence IR1 | | | | Adjusted incidence IR2 | | | |
|---|---|---|---|---|---|---|---|---|---|---|---|---|
| | Female | 95% CI | Male | 95% CI | Female | 95% CI | Male | 95% CI | Female | 95% CI | Male | 95% CI |
| 8 | 0.36 | 0.02 to 1.55 | 0.00 | 0.00 to 0.00 | 0.87 | 0.21 to 2.34 | 0.00 | 0.00 to 0.00 | 1.16 | 0.36 to 2.76 | 0.00 | 0.00 to 0.00 |
| 9 | 0.00 | 0.00 to 0.00 | 0.35 | 0.02 to 1.52 | 0.00 | 0.00 to 0.00 | 0.85 | 0.21 to 2.28 | 0.00 | 0.00 to 0.00 | 1.13 | 0.35 to 2.69 |
| 10 | 0.39 | 0.02 to 1.65 | 0.00 | 0.00 to 0.00 | 0.93 | 0.23 to 2.48 | 0.00 | 0.00 to 0.00 | 1.23 | 0.39 to 2.93 | 0.00 | 0.00 to 0.00 |
| 11 | 2.35 | 1.07 to 4.46 | 0.75 | 0.15 to 2.18 | 5.59 | 3.47 to 8.51 | 1.77 | 0.71 to 3.63 | 7.44 | 4.96 to 10.72 | 2.37 | 1.11 to 4.42 |
| 12 | 8.05 | 5.43 to 11.50 | 1.92 | 0.80 to 3.86 | 19.17 | 14.98 to 24.16 | 4.56 | 2.69 to 7.22 | 25.53 | 20.66 to 31.21 | 6.09 | 3.89 to 9.08 |
| 13 | 16.36 | 12.48 to 21.06 | 0.78 | 0.16 to 2.28 | 38.93 | 32.81 to 45.86 | 1.85 | 0.75 to 3.79 | 51.83 | 44.72 to 59.74 | 2.47 | 1.16 to 4.62 |
| 14 | 22.35 | 17.83 to 27.67 | 1.53 | 0.56 to 3.32 | 53.19 | 46.08 to 61.10 | 3.64 | 2.00 to 6.07 | 70.84 | 62.58 to 79.88 | 4.84 | 2.91 to 7.55 |
| 15 | 24.28 | 19.59 to 29.74 | 1.11 | 0.33 to 2.71 | 57.77 | 50.41 to 65.90 | 2.65 | 1.31 to 4.78 | 76.93 | 68.39 to 86.23 | 3.54 | 1.95 to 5.91 |
| 16 | 23.94 | 19.36 to 29.28 | 2.16 | 0.99 to 4.11 | 56.95 | 49.75 to 64.90 | 5.14 | 3.20 to 7.83 | 75.87 | 67.52 to 84.97 | 6.85 | 4.57 to 9.87 |
| 17 | 11.27 | 8.22 to 15.08 | 1.07 | 0.32 to 2.60 | 26.82 | 21.98 to 32.40 | 2.54 | 1.25 to 4.58 | 35.71 | 30.09 to 42.07 | 3.39 | 1.87 to 5.67 |
| Total | 10.78 | 9.77 to 11.87 | 0.96 | 0.68 to 1.30 | 25.66 | 24.09 to 27.30 | 2.28 | 1.84 to 2.79 | 34.17 | 32.36 to 36.06 | 3.03 | 2.52 to 3.62 |

IR, incidence rate.

### Web extras

Case notification definition
Please report any child/young person aged 8–17 years and 11 months inclusive, who meets the case notification definition criteria below for the first time in the last month. One bullet point criterion from each group below should be fulfilled.

Group A
► Restriction of food, low body weight, or
► Weight less than expected for age.

Group B
► Fear of gaining weight, or
► Fear of becoming fat, or
► Behaviour that interferes with weight gain, for example, excessive exercising, self-induced vomiting, use of laxatives and diuretics.

Group C
► Body image disturbance, or
► Persistent lack of recognition of the seriousness of the current low body weight.

Exclusions
► Patients who are not underweight.
► Patients with bulimia nervosa, binge eating disorder, avoidant re-strictive food intake disorder or other failure to thrive presentations.

adolescent psychiatrists in secondary care settings,[10] it is reasonable to assume that many of those cases remaining in primary care would not meet criteria for DSM5 anorexia nervosa. It is also possible that current inpatient cases are under-represented; although notifications were sent to all child and adolescent psychiatrists, including those working in inpatient settings, the main focus of the CostED study was the evaluation of community-based services, and so clinicians may have mistakenly focused on notification of community-based cases.

It must also be borne in mind that service-level (rather than population-level) incidence rates are sensitive to external factors, including service availability, funding and commissioning decisions, parental and school awareness, and stigma, all of which will impact on observed trends in incidence rates over time. The nature of community-based eating disorders services for children and adolescents in England has started to change following the publication of commissioning standards in June 2015,[25] as well as investment of £30 million to support the development of these services. The CostED incidence data were collected in 2015, 1 year before the first allocation of funding to services was made in 2016, and thus these initiatives, which may result in increases in observed incidence rates in the future, are not reflected in the data presented. Nevertheless, these estimates are approximately ten years more recent than existing secondary care data for the UK (collected between 2005 and 2006)[7] and cover a wider age range.

### Unanswered questions and future research
Future research should explore the development of earlier interventions, given evidence of an increase in incidence in young children suggesting that onset of anorexia nervosa may be starting earlier for some young

people than suggested by previous research. Research is also needed to identify approaches to the assessment of incidence simultaneously in primary and secondary care. Multinational studies should be considered for better assessment and exploration of incidence rates in young men.

## CONCLUSION

These results provide new estimates of the incidence of anorexia nervosa in young people in the UK and Ireland. While firm conclusions relating to changes in incidence rates over time for the entire sample cannot be drawn due to lack of existing secondary care evidence, service providers and commissioners should consider evidence to suggest an increase in incidence in younger children.

**Author affiliations**
[1]Health Service and Population Research Department, Institute of Psychiatry, Psychology & Neuroscience, London, UK
[2]Michael Rutter Centre for Children and Young People, South London and Maudsley NHS Foundation Trust, London, UK
[3]The Centre for Psychiatry, Imperial College London, London, UK
[4]Institute of Health Research, University of Exeter, Exeter, UK
[5]School of Social Sciences, Education & Social Work, Queen's University Belfast, Belfast, UK
[6]School for Policy Studies, University of Bristol, Bristol, UK
[7]Institute of Population Health Sciences, University of Liverpool, Liverpool, UK

**Acknowledgements** We would like to thank all the clinicians who were kind enough to give up their time to provide the CostED study with CAPSS notifications and to complete questionnaires. We know your time is precious and in short supply and we are truly grateful. We would also like to thank the team at the Child and Adolescent Psychiatry Surveillance System, based at the Royal College of Psychiatrist's Centre for Care Quality and Improvement, for their support with the design and running of the surveillance aspect of this study, and the study steering committee for their invaluable advice.

**Contributors** HP contributed to data collection, data entry, data cleaning, data analysis, and drafting of the manuscript. MS, TF, SG were co-applicants, contributed to the design of the study, provided clinical expertise, and commented on and approved the manuscript. DN was a co-applicant, contributed to the design of the study, provided expertise on CAPSS methodology, clinical expertise, and commented on and approved the manuscript. AMP contributed to the data analysis and commented on and approved the manuscript. RS contributed to data collection, data entry, data cleaning and data analysis, and commented on and approved the manuscript. NL, GK contributed to data collection, data entry and data management in Northern Ireland, and commented on and approved the manuscript. GM contributed to the design of the study, managed the Northern Ireland research staff, and commented on and approved the final report. IE was a co-applicant, contributed to the design of the study, provided clinical support to the research team, and commented on and approved the manuscript. BMB was a co-applicant, contributed to the design of the study, the data analysis, and commented on and approved the manuscript. SB was principal investigator, led the study, managed the King's College London research staff, contributed to the design of the study and the data analysis, drafted the paper, and is responsible for the overall content as guarantor.

**Funding** The study was funded by the NIHR Health Service & Delivery Research programme (11/1023/17). The study funders had no role in the study design, the collection, analysis or interpretation of data, the writing of the report, or the decision to submit the article for publication. The authors were independent from the funders and all authors had full access to all of the data in the study and can take responsibility for the integrity of the data and the accuracy of the data analysis.

**Disclaimer** The views expressed are those of the authors and not necessarily those of the NHS, the NIHR or the Department of Health and Social Care.

**Competing interests** Tamsin Ford reports she is Chair of the Child and Adolescent Psychiatry Surveillance Service that was used to run part of the study, which is an unpaid position (other than travel expenses). Kandarp Joshi reports that he was principal investigator for the Aberdeen site for a Sunovion sponsored multisite trial on effectiveness of Lurasidone in paediatric schizophrenia.

**Patient consent for publication** Not required.

**Ethics approval** The study was approved by the CAPSS Executive Committee, King's College London Research Ethics Committee (PNM/13/14-105) and the Health Research Authority (CAG 4-03(PR1)/2014) under Section 251 of the NHS Act 2006, which enables disclosure of confidential patient information, where it is not possible to use anonymised information and where seeking consent is not practical.

**Provenance and peer review** Not commissioned; externally peer reviewed.

**Data availability statement** As a result of the collection of confidential patient data without consent, and approval from the Health Research Authority for data to be provided for the purposes of the specified activity only, the data cannot be made publicly available for other purposes. However, the CostED research group will consider requests for further analysis on a case by case basis, subject to appropriate ethical/HRA approvals.

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
