## [Reviewer comments · BMJ Open]

ARTICLE DETAILS

TITLE (PROVISIONAL)	Incidence of anorexia nervosa in young people in the United Kingdom and the Republic of Ireland: A national surveillance study
AUTHORS	Petkova, Hristina; Simic, Mima; Nicholls, Dasha; Ford, Tamsin; Prina, A.Matthew; Stuart, Ruth; Livingstone, Nuala; Kelly, Grace; Macdonald, Geraldine; Eisler, Ivan; Gowers, Simon; Barrett, Barbara; Byford, Sarah

VERSION 1 – REVIEW

REVIEWER	Riittakerttu Kaltiala-Heino University of Tampere
REVIEW RETURNED	09-Nov-2018

GENERAL COMMENTS	I would like to see motivation why you chose age range 8-17. The limitation due to remarkable non-response should be emphasized still more clearly.
---

REVIEWER	On behalf of Professor Maurice Corcos (M.D., Ph.D.): Gérard Shadili (M.D.) and Aurélie Letranchant (M.D.) Professor Maurice Corcos, Doctor Gérard Shadili, Doctor Aurélie Letranchant: Adolescent and Young Adult Psychiatry Unit, Institut Mutualiste Montsouris, 42 Boulevard Jourdan, 75014, Paris, France.
REVIEW RETURNED	24-Dec-2018

GENERAL COMMENTS	The authors aimed to estimate the incidence of DSM5 anorexia nervosa in young people in contact with child and adolescent mental health services in the UK and Republic of Ireland. This study was undertaken using the Child and Adolescent Psychiatry Surveillance System (CAPSS) that has been operating since 2009 and that is based on the British Paediatric Surveillance Unit (BPSU) system. It is an observational surveillance study using the CAPSS, involving monthly reporting by child and adolescent psychiatrists between 1st February 2015 and 30th September 2015 (eight months of duration). The authors specified that the surveillance design can thus ensure a large, nationally representative sample. In addition, the presented work formed part of a study exploring the cost-effectiveness of models of care for young people with eating disorders. In the study, data were reported on young people aged 8 to 17 in contact with child and adolescent mental health services (secondary care services) for a first episode of anorexia nervosa. It's an up-to-date work. Former data were at least ten years old
--

	and most of them were derived from community-based primary care records, which fail to accurately record all new cases. Furthermore, secondary care records could be considered as a more reliable source of data on anorexia nervosa incidence than primary care registers. The authors concluded that service providers and commissioners should consider evidence to suggest an increase in incidence cases of anorexia nervosa in younger children. The submitted research is in the range of interest of the BMJ open journal. The editorial standards of the journal are respected. The compliance with ethical requirements is present. The title of the paper reflects completely the contents of the paper itself. The abstract is well-structured. In this paper, the research question is clear and the study is correctly presented. The structure of the article is accurate. The discussion is relevant. The figure and tables are appropriate. The authors detailed the STROBE checklist and thus rigorously followed these guidelines for reporting of observational studies in epidemiology. Minor comments/corrections to make:  1. Page 11, line 15: «of the positive cases» instead of «of the positive cases» 2. For the STROBE checklist, a few recommendations seem to be associated to the wrong pages. It may be due to the final structuration of the article and/or because the page 23 of the submitted article is almost blank except for the title of the figure 1. Indeed, the correct page numbers could be:  • Page 26, line 48, item 13 (a): pages «11, 23-24 » instead of «10,23» • Page 26, line 51, item 13 (b): pages «11, 23-24» instead of «10,23» • Page 26, line 52, item 13 (c): pages «23-24 » instead of «23» • Page 26, line 54, item 14 (a): page «11» instead of «10-11» • Page 27, line 4, item 16 (a): page «12» instead of «11» • Page 27, line 19, item 19: pages «13-14» instead of «12-13» • Page 27, line 22, item 20: pages «13-14» instead of «13,12 » • Page 27, line 26, item 21: pages «13-14» instead of «13» • Page 27, line 28, item 22: page «16 » instead of «15» In conclusion, after taking into account these minor corrections, this article can be accepted for publication, we don't have any other negative comments to add.
--	---

REVIEWER	Deborah Lynn Reas University of Oslo and Oslo University Hospital, Norway
REVIEW RETURNED	14-Feb-2019

GENERAL COMMENTS	This study provided useful data for service planning and prevention efforts alike. One nice methodological feature was the various safeguards on case eligibility—which is often not seen in register-based studies of incidence. As noted, there was a high proportion of unreturned cards and missing data, leading generally to less certainty about the findings and various adjustments. The authors concluded that an increase in incidence occurred in younger kids, based upon a comparison to prior studies. The authors were rightly modest in this conclusion. The data may be an artifact of methodological issues rather than a “true” increase. The period of surveillance was brief and no direct
--

	analysis of time trends across subsequent years was possible. Was this increase (whether true or artifact) observed in both males and females? Do you have enough data to specify conclusions to gender? Similar to other reports of this kind, the estimates reflect only healthcare-detected cases, which provides an underestimate of the true incidence of AN in the community. That said, a young, restricting, underweight girl with AN is arguably more likely to be detected than a normal-weight girl who binges and purges. Thus, the ascertainment bias may be less pronounced. Still, the authors should mention that most individuals with ED fail to seek or receive treatment and there are presumably many hidden cases in the community, especially as kids age and become more independent from parents/guardians. It might also be worth mentioning that this study does not address age at onset. Is there any indication that utilization of mental healthcare generally by youth has increased in the UK/Ireland? Incidence rates are sensitive to external factors. Myriad changes in service availability and financing, number of treatment providers, better awareness by schools/parents, a shifting mentality toward treatment-seeking and stigma can all affect rates. The authors are encouraged to better acknowledge such considerations in the Discussion. Finally, the authors could draw upon some literature outside of the UK. There are other studies which appear to concur with their findings. For instance, Favaro et al. (2009) doi: 10.4088/JCP.09m05176blu reported that the age of onset was decreasing in younger persons seen in their clinic between the 1980s and 2009. A recent study by Reas & Rø (2018) https://doi.org/10.1002/eat.22949 reported an increase in narrowly- and broadly-defined AN in 10-14 year girls, who seemed to "catch up" with 20-year olds in incidence between 2010 and 2016. Zerwas et al. (2015) found a peak age of incidence for females at age 15 years DOI:10.1016/j.jpsychires.2015.03.003
--	---

VERSION 1 – AUTHOR RESPONSE

Reviewer: 1

I would like to see motivation why you chose age range 8-17.

Response: The age range was based primarily on clinical knowledge of the prevalence of anorexia nervosa in young people, with the disorder being exceptionally rare in children under 8. The cut-off at 17 was due to the focus of the work on children and adolescents and the fact that most young people transition to adult services at the age of 18 in the UK. This explanation has now been added to the Inclusion and Exclusion Criteria section of the paper (page 6).

The limitation due to remarkable non-response should be emphasized still more clearly.

Response: We have now stressed that the level of non-response in the Discussion section was a 'major' constraint (page 14). However, we do not see the level of non-response as 'remarkable' given we were requesting a substantial amount of information from busy clinicians who may not have had the time or felt any investment in the study. The rate of non-response to the initial notification cards

was similar to many studies that rely on postal self-report (around 50%) and the rates of response to questionnaires were much higher than this and closer to response rates in clinical trials where people are actively pursued for interview (around 65%).

Reviewer: 2

The submitted research is in the range of interest of the BMJ open journal. The editorial standards of the journal are respected. The compliance with ethical requirements is present. The title of the paper reflects completely the contents of the paper itself. The abstract is well-structured. In this paper, the research question is clear and the study is correctly presented. The structure of the article is accurate. The discussion is relevant. The figure and tables are appropriate. The authors detailed the STROBE checklist and thus rigorously followed these guidelines for reporting of observational studies in epidemiology.

Response: We thank the reviewer for these positive comments.

Minor comments/corrections to make:

1. Page 11, line 15: «of the positive cases» instead of «of the positive cases»

Response: Thank you for spotting this. This has now been corrected.

2. For the STROBE checklist, a few recommendations seem to be associated to the wrong pages. It may be due to the final structuration of the article and/or because the page 23 of the submitted article is almost blank except for the title of the figure 1.

Response: Thank you. The STROBE checklist pages have all been checked and amended as needed.

Reviewer: 3

This study provided useful data for service planning and prevention efforts alike. One nice methodological feature was the various safeguards on case eligibility—which is often not seen in register-based studies of incidence.

Response: We thank the reviewer for these positive comments.

As noted, there was a high proportion of unreturned cards and missing data, leading generally to less certainty about the findings and various adjustments. The authors concluded that an increase in incidence occurred in younger kids, based upon a comparison to prior studies. The authors were rightly modest in this conclusion. The data may be an artifact of methodological issues rather than a “true” increase. The period of surveillance was brief and no direct analysis of time trends across subsequent years was possible. Was this increase (whether true or artifact) observed in both males and females? Do you have enough data to specify conclusions to gender?

Response: We agree with the reviewer’s assessment of these findings, hence we have tried not to overstate the conclusions. Unfortunately, the number of males in the study was too small to determine any particular patterns in the incidence rates for young men and there is little comparative data on rates for males from previous studies, and where it does exist, the rates are either not reported separately from females or the samples are similarly too small to make any meaningful comparisons. We agree this is an interesting question and have added a (perhaps rather idealistic!) research recommendation for multi-national studies to explore incidence rates in young men (page 15).

Similar to other reports of this kind, the estimates reflect only healthcare-detected cases, which provides an underestimate of the true incidence of AN in the community. That said, a young, restricting, underweight girl with AN is arguably more likely to be detected than a normal-weight girl

who binges and purges. Thus, the ascertainment bias may be less pronounced. Still, the authors should mention that most individuals with ED fail to seek or receive treatment and there are presumably many hidden cases in the community, especially as kids age and become more independent from parents/guardians. It might also be worth mentioning that this study does not address age at onset.

Response: Yes, absolutely. Fundamentally, we have included only those young people who have come to the attention of secondary-care services, so there will of course be a hidden group of young people who refuse to seek treatment and, as the reviewer notes, this is likely to increase with increasing age and independence. We have included this in the limitations, along with discussion of other 'missing' groups, such as those who come to the attention of primary care services, but we have edited slightly to emphasise the point about those who choose not to seek treatment (page 14). We have also added an additional group that may be under-represented as a result of the community-based nature of the broader CostED evaluation – those currently in inpatient services. Whilst we requested notifications from all psychiatrists, including those in inpatient settings, it is possible that the focus of the broader study led clinicians to assume that notifications of young people in inpatient services were not required (page 14).

Is there any indication that utilization of mental healthcare generally by youth has increased in the UK/Ireland? Incidence rates are sensitive to external factors. Myriad changes in service availability and financing, number of treatment providers, better awareness by schools/parents, a shifting mentality toward treatment-seeking and stigma can all affect rates. The authors are encouraged to better acknowledge such considerations in the Discussion.

Response: Yes, good point. There is indeed evidence to suggest an increasing number of referrals to CAMHS in the UK/Ireland in recent years and the recent focus in England on transforming eating disorders services for young people is likely to have an impact on rates of identification and referral – although these changes are too recent to have had much of an impact on the CostED sample. We have added discussion of these issues to the paper (pages 15).

Finally, the authors could draw upon some literature outside of the UK. There are other studies which appear to concur with their findings. For instance, Favaro et al. (2009) reported that the age of onset was decreasing in younger persons seen in their clinic between the 1980s and 2009. A recent study by Reas & Rø (2018) reported an increase in narrowly- and broadly-defined AN in 10-14 year girls, who seemed to "catch up" with 20-year olds in incidence between 2010 and 2016. Zerwas et al. (2015) found a peak age of incidence for females at age 15 years.

Response: Thank you for these useful suggestions. We have added discussion of these publications to the paper (page 14).

VERSION 2 – REVIEW

REVIEWER	Deborah Lynn Reas Oslo University Hospital Norway
REVIEW RETURNED	29-Mar-2019
GENERAL COMMENTS	The authors have addressed my concerns and I have no further comments. Despite the acknowledged limitations, this study will make a nice contribution to the literature. Well done.